# Synergistic Effects of Natural Products and Mesenchymal Stem Cells in Osteoarthritis Treatment: A Narrative Review

**DOI:** 10.3390/cimb47060445

**Published:** 2025-06-11

**Authors:** Hamoud H. Alfaqeh, Ruszymah Binti Hj Idrus, Aminuddin Bin Saim, Abid Nordin

**Affiliations:** 1School of Clinical Medicine, Chinese University of Hong Kong, Longgang District, Shenzhen 518172, China; alfaqeh@cuhk.edu.cn; 2Department of Physiology, Medical Faculty UKM, Kuala Lumpur 56000, Malaysia; ruszyidrus@gmail.com; 3Graduate School of Medicine, KPJ Healthcare University, Nilai 71800, Negeri Sembila, Malaysia; aminuddin_saim@yahoo.com; 4Ear, Nose and Throat—Head and Neck Consultant Clinic, KPJ Ampang Puteri Specialist Hospital, Ampang 68000, Selangor, Malaysia

**Keywords:** osteoarthritis, mesenchymal stem cells, natural products, curcumin, resveratrol, quercetin, epigallocatechin gallate (EGCG)

## Abstract

Osteoarthritis (OA) is a debilitating joint disorder characterized by cartilage degradation, inflammation, and loss of joint function. While mesenchymal stem cells (MSCs) hold promise for OA therapy due to their regenerative and immunomodulatory properties, challenges such as poor survival, suboptimal differentiation, and an inflammatory microenvironment limit their clinical efficacy. Natural products, including curcumin, resveratrol, quercetin, and epigallocatechin gallate (EGCG), have emerged as a complementary strategy to enhance MSC-based therapies for OA. These bioactive compounds modulate key inflammatory pathways (NF-κB, MAPK, PI3K/AKT), reduce oxidative stress, and promote chondrogenic differentiation of MSCs. Preclinical studies demonstrate the synergistic effects of MSCs and natural products in attenuating inflammation, enhancing cartilage repair, and improving joint function in OA models. However, clinical translation is hindered by challenges in bioavailability, standardization of MSC protocols, and regulatory hurdles. Future research should focus on optimizing delivery systems, conducting large-scale randomized controlled trials, and establishing personalized treatment strategies based on patient biomarkers. By addressing these challenges, the integration of natural products into MSC-based therapies could revolutionize OA treatment, offering a disease-modifying approach for millions of patients worldwide.

## 1. Introduction

Osteoarthritis (OA) is a prevalent and debilitating joint disorder characterized by the progressive degradation of cartilage, synovial inflammation, and subsequent loss of joint function. Affecting millions worldwide, OA imposes a significant socioeconomic burden, with current treatments primarily focused on symptom management rather than halting or reversing disease progression [1]. These conventional therapies (Table 1), including nonsteroidal anti-inflammatory drugs (NSAIDs) and surgical interventions, fail to address the underlying mechanisms of cartilage degradation and inflammation, highlighting the urgent need for novel therapeutic strategies [2,3].

Mesenchymal stem cells (MSCs) have emerged as a promising approach for OA treatment due to their inherent properties, including their ability to modulate immune responses, secrete trophic factors, and differentiate into chondrocytes for cartilage repair (Table 1) [4,5]. Despite these advantages, the clinical efficacy of MSC-based therapies remains inconsistent, often limited by challenges such as low cell survival, suboptimal delivery methods, and insufficient modulation of the inflammatory microenvironment in the osteoarthritic joint [6].

Natural products derived from plants, microbes, and marine organisms offer a complementary strategy to enhance the therapeutic potential of MSCs in OA [7,8,9]. Known for their diverse pharmacological properties, including anti-inflammatory, antioxidant, and cartilage-protective effects, natural products have been extensively studied in OA and regenerative medicine. Emerging evidence suggests that certain bioactive compounds, such as curcumin, resveratrol, quercetin, and epigallocatechin gallate (EGCG), can potentiate the effects of MSCs by improving cell viability, modulating differentiation, and enhancing their capacity to mitigate inflammation and promote tissue repair [10,11].

This narrative review aims to synthesize the latest findings on how natural products can significantly improve the efficacy of injected mesenchymal stem cells in treating osteoarthritis. We will discuss the mechanisms through which natural products influence MSC function, the synergistic effects observed in combination therapies, and the role of biomaterials in optimizing delivery and efficacy. Furthermore, we will address key challenges, such as standardization, bioavailability, and regulatory considerations, and highlight future research opportunities to advance this promising therapeutic paradigm.

### Literature Search Methodology

This approach ensured a balanced and up-to-date synthesis of the literature, emphasizing clinically relevant findings. This narrative review employed a comprehensive literature search across PubMed, Scopus, and Web of Science, focusing on studies published between 2018 and 2024. The search strategy utilized keywords such as “Mesenchymal stem cells” AND “osteoarthritis” AND “natural products”, “MSC therapy” AND “cartilage regeneration”, and “Curcumin OR Resveratrol OR Quercetin OR EGCG” AND “stem cell differentiation”. Inclusion criteria encompassed preclinical and clinical studies investigating the effects of natural products on MSC survival, differentiation, and OA progression, as well as review articles providing mechanistic insights into OA treatment strategies. Exclusion criteria comprised studies published before 2018, non-peer-reviewed sources, articles lacking experimental evidence, and studies focusing on non-MSC-based regenerative therapies (e.g., PRP, gene therapy). This approach ensured a balanced and comprehensive review of the current literature on the subject.

## 2. Osteoarthritis and MSC Therapies

### 2.1. Overview of Osteoarthritis (OA)

Osteoarthritis (OA) is a prevalent degenerative joint disease characterized by the deterioration of joint components, including cartilage, bone, and synovial fluid, leading to pain, stiffness, and functional limitations. It is a leading cause of disability, particularly among older adults, affecting millions of people worldwide. This condition is complex, involving mechanical wear and tear, inflammation, and cellular senescence, with no cure. Risk factors include age, female sex, obesity, genetics, and joint injuries [12]. Cellular senescence, particularly in chondrocytes and synoviocytes, contributes to the disease by producing inflammatory proteins [13]. OA is the most common joint disorder, affecting approximately 10% of men and 18% of women aged 60 years and older [14]. Its prevalence has doubled over the past decade, with significant economic and social impacts [15].

Symptoms of OA include joint pain, stiffness, swelling, and reduced mobility, which often worsen with age [16,17]. Morning stiffness and functional limitations are common, with severe cases leading to joint deformities, such as bony enlargements or misalignment, and pain at night [16,17]. Diagnosis is typically based on clinical examination and imaging techniques [17]. Treatment options include lifestyle changes, pain management, physical therapy, and surgery [16]. Pharmacological treatments often involve nonsteroidal anti-inflammatory drugs (NSAIDs) and chondroprotectors such as chondroitin sulfate. Emerging therapies include nerve growth factor inhibitors and stem cell treatments [18].

Effective management of OA requires a multidisciplinary approach that combines patient education, pharmaceuticals, and complementary therapies [19]. Early diagnosis, prevention, and tailoring of treatment plans are crucial to achieving optimal outcomes. Person-centered care focusing on modifiable factors, such as exercise and weight loss, is recommended [12]. While osteoarthritis remains a significant health challenge, ongoing research into its pathophysiology and treatment options offers hope for improved management strategies.

With the global rise in aging populations and obesity rates, the prevalence of OA is expected to increase, further exacerbating the societal and economic burden associated with this disease. Current treatments for OA primarily focus on symptom management through pharmacological interventions, such as NSAIDs, corticosteroids, and hyaluronic acid injections, alongside physical therapy and, in severe cases, joint replacement surgery [20]. However, these strategies do not address the underlying pathophysiology of OA, and their long-term efficacy is often limited [3]. This gap in therapeutic options has driven interest in regenerative medicine approaches that aim to restore joint structure and function.

### 2.2. Pathophysiology of Osteoarthritis

Osteoarthritis (OA) is a complex, multifactorial joint disorder characterized by the degeneration of articular cartilage, synovial inflammation, and subchondral bone changes. OA pathophysiology involves biomechanical, biochemical, and cellular factors contributing to joint degradation and pain.

Recent research has provided insights into the molecular and cellular mechanisms underlying OA, highlighting the roles of inflammation, cartilage metabolism, and bone remodeling. Cartilage degradation is a hallmark of OA, primarily due to an imbalance between the anabolic and catabolic processes within the joint. Biochemical markers such as collagen type II cleavage products (C2C) and cartilage oligomeric matrix protein (COMP) are elevated in OA, indicating cartilage degradation [21]. Mechanical and immune-mediated injuries contribute to cartilage destruction, with inflammatory cytokines such as interleukin-1β (IL-1β) and tumor necrosis factor-alpha (TNF-α) playing significant roles [22].

Chronic low-grade synovial membrane inflammation is a central feature of OA and is driven by innate and adaptive immune responses [23]. Inflammatory cytokines, including IL-6, are elevated in the synovial fluid of OA patients, correlating with pain and disease progression [21,24]. MicroRNAs (miRNAs) regulate inflammatory processes and cartilage metabolism, influencing OA progression [25]. Subchondral bone lesions often precede cartilage degeneration, suggesting they play a pivotal role in OA pathogenesis [23]. Mechanical loading and inflammatory stimuli affect osteocyte activity, leading to bone remodeling and the osteophyte formation [24]. Understanding the interplay between bone and cartilage changes is crucial for understanding OA progression and developing therapeutic strategies [26].

Chondrocyte apoptosis and other forms of programmed cell death, such as PANoptosis, contribute to cartilage degradation in OA [27]. These cellular processes are influenced by mechanical stress and inflammatory mediators, highlighting the complexity of the OA pathophysiology (Figure 1) [27]. Current treatments focus on symptom management; however, novel therapies target underlying pathophysiological mechanisms. Native collagen type II and Aflapin have shown promise in slowing OA progression by modulating immune responses and cartilage metabolism [22]. Regenerative approaches, including cell and gene therapies, have been explored to promote cartilage repair and joint functions [26]. While the pathophysiology of osteoarthritis is increasingly understood, challenges remain in translating this knowledge into effective treatments. The heterogeneity of OA among patients complicates the development of universal therapies and necessitates personalized approaches. Additionally, neuroinflammation and central sensitization in pain chronicity present further complexities for managing OA symptoms [23,28].

### 2.3. Inflammatory Pathways in Osteoarthritis

Inflammatory pathways play a crucial role in the pathogenesis and progression of osteoarthritis (OA), a degenerative joint disease characterized by cartilage degradation, synovial inflammation, and subchondral bone changes. These pathways involve a complex interplay of cytokines, signaling molecules, and cellular responses that contribute to the chronic nature of the disease.

Cytokines such as interleukin-1 beta (IL-1β), IL-6, IL-17, and TNF-α drive inflammation and cartilage degradation (Figure 2). These cytokines activate signaling pathways such as NF-κB and MAPK, which produce MMPs that degrade the extracellular matrix [29,30,31]. Synovial macrophages and synoviocytes release inflammatory cytokines that perpetuate synovitis and joint damage. The TNF and chemokine signaling pathways are particularly significant in synovial inflammation [32,33].

Oxidative stress is closely linked to inflammation in OA, with reactive oxygen species (ROS) and nitric oxide (NO) contributing to cartilage damage and inflammation. The Nrf2 signaling pathway, which regulates oxidative stress responses, interacts with NF-κB and influences inflammation and cellular metabolism [34,35]. Several other signaling pathways have been implicated in OA, including the Wnt, Notch, and TGF-β pathways, which regulate chondrocyte apoptosis, cartilage matrix synthesis, and degradation [30,36]. MAPK and toll-like receptor (TLR) pathways are critical for mediating inflammatory responses and promoting the release of inflammatory mediators, thereby exacerbating cartilage destruction [30,36]. In post-traumatic osteoarthritis (PTOA), inflammation is triggered by joint trauma, which leads to mitochondrial dysfunction and cytokine release. Calcium/calmodulin-dependent protein kinase kinase 2 (CaMKK2) has been identified as a critical regulator of inflammatory responses and a potential therapeutic target [36,37].

Targeting the inflammatory pathways offers promising therapeutic avenues for OA treatment. Modifying the pro-inflammatory and anti-inflammatory prostaglandin balance could help cartilage regeneration and reduce inflammation [38]. The recruitment of mesenchymal stem cells and the transition from pro-inflammatory to anti-inflammatory states are potential strategies for developing new OA treatments [38]. Natural compounds targeting oxidative stress pathways have shown potential in reducing inflammation and protecting cartilage, highlighting the therapeutic potential of modulating oxidative stress in OA [34]. Although inflammation is a central component of OA pathogenesis, it is essential to consider the multifactorial nature of the disease, including mechanical stress and genetic predisposition, when developing comprehensive treatment strategies. Understanding the complex inflammatory pathways in OA highlights the need for diverse treatment strategies, including the exploration of current pharmacological interventions, as detailed in the next section.

### 2.4. Mesenchymal Stem Cells (MSCs)

Mesenchymal stem cells (MSCs) have garnered significant attention as a promising regenerative therapy for OA [39]. MSCs are multipotent progenitor cells that differentiate into mesenchymal lineages, including chondrocytes, osteoblasts, and adipocytes. They also possess potent immunomodulatory and anti-inflammatory properties, which are particularly relevant for the treatment of OA [40,41]. By secreting bioactive molecules such as cytokines, growth factors, and extracellular vesicles, MSCs can modulate the inflammatory microenvironment, inhibit the catabolic activity of inflammatory cytokines, and promote cartilage repair. Despite their potential, the clinical application of MSCs in OA therapy faces several challenges. These include low survival rates following intra-articular injection, poor engraftment and retention within the joint, and limited capacity to overcome the hostile inflammatory microenvironment in osteoarthritic joints [42]. Additionally, variations in MSC source, isolation techniques, and culture conditions have contributed to inconsistent therapeutic outcomes in preclinical and clinical studies [6,43]. Addressing these limitations requires innovative strategies to enhance the therapeutic efficacy of MSCs.

Recent studies have highlighted the potential of combining MSC-based therapies with natural products to overcome these barriers [44,45]. Natural products can modulate the inflammatory and oxidative stress pathways implicated in OA pathogenesis while enhancing MSC viability, functionality, and differentiation [46,47]. These synergistic effects offer a compelling rationale for integrating natural products into MSC-based therapeutic regimens for OA. This integration could potentially enhance the therapeutic efficacy of MSCs, leading to improved outcomes in patients suffering from osteoarthritis. Additionally, incorporating natural compounds may help modulate the inflammatory response and promote tissue regeneration, thereby addressing the disease’s underlying pathology.

## 3. Natural Products

### 3.1. Natural Products in Enhancing MSC Therapies

Natural products derived from diverse biological sources, such as plants, microbes, and marine organisms, have long been recognized for their pharmacological properties in treating various diseases [48,49,50]. Their bioactive compounds exhibit a wide range of effects, including anti-inflammatory, antioxidant, and chondroprotective activities, which are particularly relevant to treating osteoarthritis (OA) [34,46,51]. In recent years, there has been growing interest in leveraging natural products to enhance the therapeutic potential of mesenchymal stem cells (MSCs) in OA management [52]. This trend is driven by the increasing evidence suggesting that natural compounds possess unique properties supporting MSC proliferation, differentiation, and overall functionality. Researchers have begun investigating various natural substances, including plant extracts, vitamins, and minerals, to determine their synergistic effects when combined with MSC therapies [53].

The pathophysiology of OA marks an imbalance between pro-inflammatory and anti-inflammatory mediators, oxidative stress, and extracellular matrix degradation [54]. While MSCs exhibit significant potential in modulating these pathological processes, their therapeutic efficacy can be hindered by low viability and limited functionality within the hostile inflammatory microenvironment of osteoarthritic joints. Natural products can address these challenges by modulating key signaling pathways and creating a more favorable environment for MSC-based therapies [45]. By enhancing the viability and functionality of injected MSCs, these natural products can lead to improved outcomes in osteoarthritis management. This enhancement can be attributed to the synergistic effects of specific natural compounds that promote cellular proliferation and mitigate senescence [55]. Recent studies have highlighted various natural products, such as curcumin, resveratrol, and certain flavonoids, that have significantly improved MSC characteristics under inflammatory conditions typical of osteoarthritis [56,57,58,59].

Several natural products have demonstrated the ability to enhance MSC survival, proliferation, and differentiation [53]. For instance, curcumin, a polyphenol derived from turmeric, has been shown to exert strong anti-inflammatory and antioxidant effects, reducing the expression of pro-inflammatory cytokines such as interleukin-1β (IL-1β) and tumor necrosis factor-alpha (TNF-α) [54,60]. Similarly, resveratrol, a stilbene found in grapes and red wine, promotes MSC proliferation and differentiation into chondrocytes while mitigating oxidative stress and inflammatory signaling pathways [61]. Natural products can also enhance MSC functionality and immunomodulatory properties. Quercetin, a flavonoid commonly found in fruits and vegetables, has been reported to reduce the levels of matrix metalloproteinases (MMPs) that degrade cartilage and modulate the polarization of macrophages toward an anti-inflammatory phenotype [47,52,62]. Furthermore, epigallocatechin gallate (EGCG), which is recognized as a major catechin found in green tea, has been shown through various studies to protect mesenchymal stem cells (MSCs) from apoptosis [63,64,65,66]. This beneficial compound also enhances their differentiation potential significantly and plays a crucial role in reducing the production of various inflammatory mediators that can affect cell health.

The integration of natural products into MSC therapies offers several advantages. These compounds can precondition MSCs in vitro, enhancing their viability and therapeutic properties before administration. Alternatively, they can be co-administered with MSCs to create a synergistic effect within the joint microenvironment. However, the efficacy of these approaches depends on factors such as the bioavailability, stability, and dosage of the natural products, which remain critical challenges to address. The potential of natural products to enhance MSC-based therapies for OA represents an exciting frontier in regenerative medicine. Combining the regenerative capabilities of MSCs with the pharmacological benefits of natural products may allow for the development of more effective and targeted therapies for OA patients.

A growing body of evidence supports the role of natural products in improving the efficacy of MSC-based therapies for osteoarthritis. Table 2 summarizes key studies evaluating the effects of various bioactive compounds on MSC survival, differentiation, and cartilage repair in preclinical and clinical models.

### 3.2. Specific Natural Products and Their Effects on MSCs

Integrating bioactive natural products can significantly enhance the therapeutic potential of mesenchymal stem cells (MSCs) in osteoarthritis (OA). These compounds, derived from plants and other natural sources, possess unique properties that address key challenges associated with MSC-based therapies, such as limited cell survival, suboptimal differentiation, and inadequate immunomodulation. Below, we discuss the effects of several well-studied natural products and their mechanisms of action in augmenting MSC function for OA treatment.

#### 3.2.1. Curcumin

Curcumin, a polyphenolic compound derived from the turmeric root (Curcuma longa), is renowned for its potent anti-inflammatory and antioxidant properties [76,77]. It inhibits key inflammatory pathways, such as nuclear factor-kappa B (NF-κB), and reduces the production of pro-inflammatory cytokines like interleukin-6 (IL-6) and tumor necrosis factor-alpha (TNF-α). Studies have demonstrated that curcumin enhances MSC survival and proliferation under oxidative stress conditions, making it a valuable agent for preconditioning MSCs before their therapeutic application [78,79,80,81]. Additionally, curcumin has been shown to promote the chondrogenic differentiation of mesenchymal stem cells (MSCs) [82,83], thereby facilitating and enhancing cartilage repair in various osteoarthritis (OA) models. This process is crucial for improving joint function and reducing pain in affected individuals.

#### 3.2.2. Resveratrol

Resveratrol, a stilbene found in grapes, red wine, and berries, has garnered attention for its antioxidative and chondroprotective effects. It activates the sirtuin-1 (SIRT1) signaling pathway, which regulates mitochondrial function and reduces oxidative damage in MSCs [72]. Resveratrol has also been shown to enhance MSC-mediated cartilage repair by promoting the expression of cartilage-specific markers, such as collagen type II and aggrecan [84]. Furthermore, resveratrol suppresses matrix metalloproteinases (MMPs) and inflammatory mediators, contributing to the preservation of cartilage integrity in osteoarthritic joints [61,71,85]. This action not only aids in significantly reducing inflammation but also actively promotes the regeneration of chondrocytes, which are essential for maintaining the structural integrity of cartilage. Furthermore, various studies have clearly shown that combining resveratrol with mesenchymal stem cells may notably enhance the therapeutic effects [68,69,78], ultimately leading to improved outcomes in osteoarthritis treatment, making it a promising avenue for further research and application in clinical settings.

#### 3.2.3. Quercetin

Quercetin is a flavonoid abundant in fruits, vegetables, and teas. Its anti-inflammatory and immunomodulatory properties make it an attractive candidate for MSC-based therapies. Quercetin reduces the activity of MMPs, which are involved in cartilage degradation, and modulates the polarization of macrophages in the joint microenvironment [73,86]. This shifts the inflammatory profile toward a reparative state, enhancing the therapeutic effects of MSCs. Quercetin has also been shown to promote MSC differentiation into chondrocytes, further supporting cartilage regeneration in OA [87]. This specific process significantly enhances the therapeutic potential of mesenchymal stem cells (MSCs) and may ultimately lead to improved clinical outcomes for individuals suffering from osteoarthritis.

#### 3.2.4. Epigallocatechin Gallate (EGCG)

EGCG, the primary catechin in green tea, is recognized for protecting MSCs from apoptosis and oxidative stress. This protective effect is crucial in osteoarthritis, where oxidative stress plays a significant role in the degeneration of cartilage and the overall pathology of the disease [81]. Recent studies have demonstrated that the application of EGCG enhances the viability of MSCs and promotes their regenerative properties, leading to improved outcomes in joint health [75]. By reducing reactive oxygen species (ROS) levels and inhibiting the activation of NF-κB, EGCG enhances MSC viability and function. Additionally, EGCG has been shown to improve the chondrogenic potential of MSCs by upregulating the expression of chondrocyte-specific genes [88]. Its significant ability to effectively suppress pro-inflammatory cytokines and various chemokines further strengthens and supports its potential use in MSC-based therapies designed specifically for osteoarthritis (OA).

Natural products combined with MSCs create synergistic effects that enhance therapeutic potential. Preconditioning MSCs with curcumin or EGCG improves their survival and regenerative capacity [74,89], while resveratrol or quercetin can modulate inflammation in osteoarthritic joints [69,86]. These combinations address the limitations of standalone MSC therapies, yet clinical translation encounters challenges with bioavailability, stability, and delivery methods. Innovative strategies, such as nanoparticle encapsulation or hydrogels, should be explored to optimize efficacy in MSC applications. Advanced delivery methods could also enhance the retention and viability of MSCs in osteoarthritic environments, improving their regenerative capabilities and tissue repair. These advancements may facilitate the delivery of bioactive compounds to modulate inflammation, supporting MSC therapeutic effects in osteoarthritis. Integrating natural products improves MSC viability and differentiation, contributing to effective treatment strategies [55]. Furthermore, these compounds enhance immunomodulatory properties, promoting better MSC integration within the osteoarthritic microenvironment and boosting their regenerative capacity [88]. Thus, natural products are vital for enhancing MSC therapeutic efficacy and patient outcomes in osteoarthritis. This section outlines how natural products enhance MSC function, paving the way for integrated therapeutic approaches for OA. Future research should optimize these combinations and identify additional natural compounds with complementary effects.

## 4. Combination Therapies: MSCs and Natural Products

Combining mesenchymal stem cells (MSCs) with natural products represents a novel and promising approach to treating osteoarthritis (OA). This strategy leverages the regenerative and immunomodulatory properties of MSCs alongside the bioactive potential of natural compounds, creating synergistic effects that can enhance therapeutic outcomes. Here, we explore the evidence supporting these combination therapies, focusing on their synergistic effects, preclinical and clinical studies, and the mechanisms underlying their enhanced efficacy.

### 4.1. Synergistic Effects of MSCs and Natural Products

The synergistic effects of mesenchymal stem cells (MSCs) and natural products, such as curcumin, resveratrol, quercetin, and epigallocatechin gallate (EGCG), have been shown to enhance MSC function by addressing limitations like poor cell survival, inadequate differentiation, and insufficient modulation of the inflammatory microenvironment (Table 3). These natural compounds, known for their antioxidant and anti-inflammatory properties, interact with MSCs to improve their therapeutic efficacy in various pathological conditions. The following sections explore the mechanisms through which these natural products enhance MSC function.

Curcumin, a polyphenolic compound, has improved MSC survival by reducing oxidative stress and inflammation. It interacts with key proteins involved in inflammation and cancer progression, such as COX-2 and NF-κB, enhancing MSC viability and differentiation potential [79,81,90]. Resveratrol promotes osteogenic differentiation of MSCs by modulating the NRF2/HO-1 and NF-κB pathways, which are crucial for reducing inflammation and enhancing bone regeneration [91]. Quercetin, another polyphenol, has been found to synergize with other compounds to promote osteogenesis and reduce inflammation in MSCs, particularly in the context of age-related bone loss [69,92,93]. Curcumin and resveratrol have demonstrated a synergistic ability to inhibit TNF-α-induced inflammation by suppressing the NF-κB signaling pathway, which is vital for reducing vascular inflammation and enhancing the function of mesenchymal stem cells (MSCs) in inflammatory conditions [94,95,96]. Additionally, quercetin, with its strong antioxidant properties, modulates signaling pathways that influence inflammation, thereby strengthening the immunomodulatory capacity of MSCs [97,98].

Despite curcumin’s low bioavailability, its therapeutic efficacy can be improved through structural modifications or nano-based delivery systems. These approaches enhance solubility and absorption, maximizing its potential when used with MSCs [99,100,101,102]. Notably, the combination of resveratrol and curcumin at low concentrations has been observed to produce a synergistic effect, which is more effective than the individual compounds alone, thereby improving the therapeutic outcomes of MSC-based therapies [103,104,105,106]. Research has also shown that MSCs preconditioned with natural antioxidants, such as crocin, significantly improve cell survival and reduce oxidative stress. This highlights the promising role of natural products in enhancing MSC efficacy, particularly in lung injury models [1,107]. Furthermore, resveratrol has been found to alleviate inflammation and promote osteogenic differentiation in periodontal tissues, showcasing its potential applications in dental MSC therapies [108].

**Table 3 cimb-47-00445-t003:** The synergistic effects of mesenchymal stem cells (MSCs) and natural products.

Category	Natural Product	Model Used	Outcomes	Author
Enhancement of MSC Survival and Differentiation	Curcumin Dosage regimens: General wellness: 500–1000 mg daily. Arthritis and joint pain: 1000–1500 mg daily, often divided into smaller doses. Heart health: 500–700 mg daily. Cognitive health: 500–1000 mg daily. Maximum safe limit: up to 8000 mg per day for short durations.	in vitro/ clinical studies	Decreased oxidative stress and inflammation improve MSC viability and differentiation potential.	[79,81,90]
	Resveratrol Dosage regimens: Anti-aging: 150–500 mg per day; up to 1000 mg daily for more pronounced effects. Heart health: 100–250 mg daily; higher doses of 250–500 mg per day for existing cardiovascular conditions. Anti-inflammatory: up to 1500 mg daily for up to 3 months; higher doses of 2000–3000 mg daily for 2–6 months.	in vitro	Promotes bone growth, reduces inflammation, and enhances regeneration.	[69,78,108]
	Quercetin Dosage regimens: General wellness: 500 mg per day, often combined with vitamin C or bromelain to enhance absorption. Allergies or inflammation: 500–1000 mg per day, divided into multiple doses.	in vitro	Supports bone formation and decreases inflammation, especially in age-related bone loss.	[69,92,93]
Modulation of Inflammatory Microenvironment	Curcumin and resveratrol combination (no information)	in vitro/in vivo	Inhibits TNF-α-induced inflammation, suppresses NF-κB signaling, reduces vascular inflammation, and enhances MSC function in inflammatory conditions.	[95,96,109,110]
	Quercetin Dosage regimens: General wellness: 500 mg per day, often combined with vitamin C or bromelain to enhance absorption. Allergies or inflammation: 500–1000 mg per day, divided into multiple doses.	in vitro/in vivo	Enhances MSCs’ immunomodulatory capacity and modulates inflammatory signaling pathways.	[97,98]
Improvement of Bioavailability and Efficacy	Curcumin Dosage regimens: General wellness: 500–1000 mg daily. Arthritis and joint pain: 1000–1500 mg daily, often divided into smaller doses. Heart health: 500–700 mg daily. Cognitive health: 500–1000 mg daily. Maximum safe limit: up to 8000 mg per day for short durations	in vitro/in vivo	Improved efficacy through structural modifications or nano-delivery systems enhances solubility and absorption, maximizing the therapeutic potential of MSCs.	[99,100,101,102]
	Resveratrol and curcumin (no information)	in vitro/in vivo	Low concentrations of compounds work synergistically, improving the therapeutic outcomes of MSC-based therapies more effectively than when used individually.	[103,104,105,106]
Case Studies and Applications	Crocin (no information)	in vivo	Significant improvements in cell survival and reductions in oxidative stress highlight the potential of natural products to enhance MSC efficacy in lung injury models.	[1,107]
	Resveratrol Dosage regimens: Anti-aging: 150–500 mg per day; up to 1000 mg daily for more pronounced effects. Heart health: 100–250 mg daily; higher doses of 250–500 mg per day for existing cardiovascular conditions. Anti-inflammatory: up to 1500 mg daily for up to 3 months; higher doses of 2000–3000 mg daily for 2–6 months	in vivo	Ameliorates inflammation, promotes osteogenic differentiation in periodontal tissues, and demonstrates potential in dental applications of MSC therapy.	[108]

Integrating mesenchymal stem cells (MSCs) with natural products like curcumin and resveratrol offers promise in regenerative medicine, but low bioavailability remains a key challenge. While these compounds are identified for enhancing MSC functionality, their clinical application is hindered by poor solubility and absorption [69,111,112]. Nano-formulations and structural modifications may improve bioavailability, yet current research has not sufficiently explored these strategies. Curcumin is known for its anti-inflammatory and potential anti-cancer effects, but a lack of substantial research on its safety and absorption limits its clinical use. Effective dosing regimens need to be established for practical applications. Similarly, the role of resveratrol in inflammation, particularly regarding bone regeneration, requires further investigation to understand its long-term effects. The absence of rigorous clinical trials for natural compounds like the BlastiMin Complex exacerbates the situation. While some studies suggest that curcumin and resveratrol may work synergistically to reduce inflammation, their long-term effects on diverse populations warrant more research [113]. Additionally, there are challenges in achieving optimal blood levels of these phytochemicals, emphasizing the need for better absorption methods. The interaction between crocin and dexamethasone in mitigating lung cell injuries raises questions about long-term MSC survival and functionality [114], which remain unexplored. In conclusion, while the combination of MSCs and natural products holds significant potential, critical gaps in research concerning absorption, safety, and long-term effects must be addressed to enhance clinical applications and improve patient outcomes.

### 4.2. Evidence from Preclinical Studies

Preclinical research has demonstrated that applying mesenchymal stem cells (MSCs) alongside natural products can exhibit significant potential in treating osteoarthritis. These investigations indicate that including natural compounds, particularly those derived from various plants, enhances the proliferation and differentiation of MSCs, facilitating cartilage repair and mitigating inflammation.

For instance, the administration of curcumin-treated MSCs into the joint has shown effectiveness in osteoarthritis models by diminishing inflammation and promoting cartilage regeneration more effectively than using MSCs alone [115]. Curcumin operates by inhibiting the NF-κB/JNK signaling pathway, which is crucial in the progression of osteoarthritis [81,116]. Innovative delivery systems, such as curcumin-loaded nanoparticles and microgels, have improved the bioavailability and efficacy of these treatments, resulting in reduced joint swelling and slower cartilage degradation [117,118]. Furthermore, curcumin provides protective benefits to cartilage cells and aids cartilage repair by modulating various biological pathways while decreasing oxidative stress and cell apoptosis [88,119]. Curcumin’s efficacy in addressing cartilage damage is further enhanced when combined with other substances like chondroitin sulfate [120]. In conclusion, curcumin, particularly when integrated with advanced delivery techniques or MSCs, presents a promising method for augmenting cartilage repair and diminishing inflammation in osteoarthritis.

Nonetheless, some limitations and challenges involve the possibility of differing bioavailability and the necessity for standardized formulations. Additionally, while the initial findings are encouraging, further clinical trials are essential to validate the efficacy and safety of these combinations in a broader patient population. This is particularly critical given the complex nature of osteoarthritis and the varying responses observed among different demographic groups. Future studies should optimize the dosages and combinations of these natural products to ensure maximum therapeutic benefit and minimal adverse effects.

In addition to curcumin, resveratrol and MSCs also demonstrate the potential to reduce cartilage damage and oxidative stress in osteoarthritis animal models [71,83,85]. Resveratrol is recognized for its anti-inflammatory and antioxidant properties, playing a significant role in managing osteoarthritis, a condition characterized by joint inflammation and cartilage degradation [85]. This natural compound influences several vital biological pathways, offering protection to cartilage and enhancing the effectiveness of MSC therapies [61,121]. MSCs, especially those isolated from adipose or joint tissues, display robust regenerative properties that help reduce oxidative stress and inflammation [122]. Combining resveratrol and MSCs may enhance cellular survival and proliferation [61,72]. Research suggests that resveratrol activates SIRT1, a critical factor in regulating oxidative stress and inflammation, thereby supporting its potential synergistic effects with MSCs in decelerating osteoarthritis progression [103,123,124]. Additionally, resveratrol can inhibit detrimental inflammatory proteins, improving MSC-based interventions [125].

The combined effects of resveratrol and mesenchymal stem cells (MSCs) indicate a potential avenue for advancing more effective treatments for osteoarthritis that target cartilage damage and oxidative stress. Nevertheless, additional research is necessary to identify the most effective dosages and combinations that enhance therapeutic benefits while reducing possible adverse effects. Moreover, it is important to comprehend how these natural products improve MSC effectiveness, as this knowledge will be vital for creating targeted treatment strategies for osteoarthritis patients. This insight could facilitate the formulation of synergistic therapies that enhance the regenerative capabilities of MSCs, ultimately benefiting clinical outcomes for individuals afflicted with osteoarthritis.

Quercetin, a flavonoid with significant health-promoting properties, has been shown to enhance mesenchymal stem cells’ survival and anti-inflammatory effects, thereby improving joint health [126]. Studies have demonstrated that quercetin can ameliorate senescence and promote osteogenesis in bone marrow mesenchymal stem cells (BMSCs) by inhibiting the repetitive element-triggered RNA sensing pathway, which is crucial for maintaining the osteogenic capacity of BMSCs [127,128,129]. Additionally, quercetin-treated BMSCs produce exosomes that deliver miR-124-3p to chondrocytes, inhibiting osteoarthritis progression by reversing the apoptotic effects of inflammatory cytokines and modulating key signaling pathways such as MAPK/p38 and NF-κB [130].

Quercetin also exhibits potent anti-inflammatory and antioxidant properties, which are essential in reducing the expression of pro-inflammatory cytokines and mitigating oxidative stress, both of which are critical in the pathogenesis of osteoarthritis and other inflammatory joint diseases [86,131,132]. Furthermore, quercetin has been shown to prevent osteoarthritis progression by maintaining the balance of inflammatory cascades and promoting cartilage repair, as evidenced by its ability to inhibit matrix metalloproteinases and enhance the expression of cartilage-protective proteins [70,73,86,133,134].

These findings collectively highlight the potential of quercetin as a therapeutic agent in enhancing MSC survival and function, offering a promising approach to improving joint health and managing conditions like osteoarthritis. However, integrating natural products like quercetin into MSC therapies requires further investigation. Future research should focus on elucidating the mechanisms by which quercetin influences stem cell behavior and conducting clinical trials to assess its efficacy in osteoarthritis patients. Moreover, exploring synergistic effects with other natural compounds that may further enhance the therapeutic potential of mesenchymal stem cells in osteoarthritis treatment is crucial.

Epigallocatechin-3-gallate (EGCG) has been shown to play a significant role in protecting mesenchymal stem cells (MSCs) from inflammation and enhancing their chondrogenic differentiation both in vitro and in vivo [135,136]. EGCG, a polyphenolic antioxidant, has been reported to support MSC-mediated collagen remodeling under oxidative stress conditions; however, it is ineffective when the collagen has already undergone oxidative damage [137]. Instead, EGCG facilitates alternative routes for collagen removal, such as internalization and transcytosis, which are crucial for maintaining ECM integrity [136,137].

In the context of arthritis, EGCG combined with extracellular vesicles has demonstrated a significant upregulation of type II collagen expression, which is essential for cartilage repair, and a reduction in apoptosis-related proteins, thereby ameliorating cartilage destruction in rheumatoid arthritis models [71,138]. Furthermore, EGCG has been shown to increase type II collagen accumulation in MSC-derived cartilage-like sheets by suppressing collagen degradation, highlighting its potential to enhance chondrogenic differentiation [139]. Additionally, EGCG’s anti-inflammatory properties have been observed to reduce cartilage inflammation and degradation in osteoarthritis models, suggesting its role as a disease-modifying agent [140]. The protective effects of EGCG extend to hypoxia-induced conditions, where it mitigates apoptosis and promotes osteogenic differentiation in MSCs by upregulating miR-210, which downregulates EFNA3, a receptor tyrosine kinase ligand [94,141].

These findings highlight EGCG’s multifaceted role in supporting MSC function and differentiation, making it a promising candidate for enhancing MSC-based therapies for cartilage regeneration and protection against inflammatory insults. Nonetheless, it is important to consider the underlying mechanisms through which EGCG operates. By enhancing the proliferative capacity of MSCs and promoting their migration to damaged tissues, EGCG not only aids in cartilage repair but also serves as an anti-inflammatory agent. Future studies should focus on optimizing the delivery methods of EGCG in conjunction with MSC injections to maximize therapeutic outcomes.

### 4.3. Insights from Clinical Trials

The clinical evidence regarding using mesenchymal stem cell (MSC) and natural product combination therapies in patients with osteoarthritis (OA) shows promise, though it is still in its early stages. MSCs have garnered significant attention for their potential to treat OA, largely owing to their ability to differentiate into various tissues and their strong paracrine effects, which include anti-inflammatory and immunomodulatory properties [142,143]. Clinical trials have demonstrated that MSCs can alleviate pain and enhance joint functionality, with various studies indicating favorable outcomes regarding cartilage protection and repair [144,145,146].

The exploration of combining MSCs with natural products, such as Radix Achyranthis Bidentatae (AB), has emerged as a compelling area of interest. AB is recognized for its potential to improve the internal conditions of knee joints and stimulate MSC repair mechanisms, suggesting a synergistic effect when used alongside MSCs [147,148]. Natural products generally possess anti-inflammatory and antioxidant properties that may further enhance the therapeutic effects of MSCs [149,150]. Despite these encouraging findings, the available clinical evidence remains limited, with most studies concentrating on preclinical models or small-scale clinical trials. For instance, a meta-analysis of MSCs combined with platelet-rich plasma (PRP) indicated improvements in pain and joint function [151]; however, the particular role of natural products in these combinations necessitates further investigation [152,153,154].

The complexities surrounding MSC metabolism, coupled with the differences in study designs and patient characteristics, create challenges in reaching definitive conclusions regarding the effectiveness of MSC–natural product therapies [155,156,157]. As a result, despite the promising possibilities offered by combining MSC and natural products for treating osteoarthritis, it has become evident that there is an urgent requirement for carefully structured clinical trials. Such trials are essential for a comprehensive assessment of both the safety and efficacy of these innovative therapies [158,159].

Moreover, the current clinical trials examining MSC–natural product combination therapies for OA face numerous challenges stemming from the complexities inherent to both MSC and natural product treatments. A key hurdle is the variability in the microenvironment of osteoarthritic joints, which can impact MSC efficacy. Factors such as inflammation, senescence, hypoxia, high glucose levels, and elevated lipid content in the joint environment can adversely affect MSC function, complicating their therapeutic potential [160]. Additionally, the clinical translation of MSC therapies is impeded by the variety of cell sources, isolation techniques, and culture protocols, which hinders comparisons across studies and the establishment of standardized treatment protocols [45].

Although MSCs have demonstrated promise in preclinical models, their effectiveness in human trials has not been consistent, with some studies revealing only short-term benefits and others presenting contradictory findings [161,162]. The challenges related to natural products include issues with bioavailability, precise targeting, and the heterogeneity of OA as a disease, complicating the formulation of effective natural product-based therapies [163,164,165]. Furthermore, the current clinical trial evidence is insufficient to definitively ascertain the efficacy of natural anti-inflammatory products in preventing or treating OA, highlighting the need for additional high-quality research to explore their mechanisms and optimal delivery systems [166].

The integration of MSCs with natural products in clinical trials is also challenged by the necessity for robust methodologies to evaluate the combined effects and identify specific patient sub-populations that might derive the greatest benefit from these therapies [167]. In summary, while the potential for MSC–natural product combination therapies in OA is hopeful, substantial research and methodological advancements are essential to overcome these challenges and realize reliable clinical outcomes.

### 4.4. Mechanisms Underlying Synergistic Effects

Natural products play a crucial role in reducing oxidative stress by decreasing levels of reactive oxygen species (ROS) and safeguarding mesenchymal stem cells (MSCs) and the joint microenvironment from harm. The beneficial effects of these products stem from key antioxidant pathways and the interactions among various natural compounds. For example, botanical extracts such as turmeric, pagoda tree seed pod, and rosemary activate the Nrf2-ARE signaling pathway, enhancing the body’s defense mechanisms against free radicals and thus lowering oxidative stress [168,169,170,171]. Additionally, polyphenols and glutathione (GSH) promote cellular redox homeostasis, demonstrating synergistic effects that support disease management [172,173,174].

In the context of MSC therapy, molecular hydrogen and cold atmospheric plasma have been shown to reduce ROS levels and activate the Nrf2 pathway, thereby improving MSC survival and functionality [175,176]. Further, combinations of natural compounds such as α-tocopherol, ascorbic acid, and selenium offer protective benefits against chondrocyte cell death, presenting a promising strategy to combat oxidative stress in joint health [177]. A supplement formula containing vitamin D, Lactobacillus rhamnosus, ginger, curcumin, and Boswellia extract effectively reduces oxidative stress and promotes MSC differentiation, indicating a synergistic effect [178]. These findings highlight the significance of leveraging the synergistic effects of natural products to alleviate oxidative stress and enhance MSC therapy in regenerative medicine.

Moreover, compounds like curcumin and quercetin not only exhibit anti-inflammatory properties but also modulate pathways such as NF-κB and IL-6. Curcumin, derived from *Curcuma longa* L., enhances the efficacy of chemotherapeutic drugs by regulating inflammatory cancer pathways [179,180]. When paired with other compounds such as resveratrol, curcumin displays synergistic effects in inflammatory reduction by inhibiting NF-κB signaling, which is critical for addressing endothelial dysfunction and cardiovascular diseases [109,181,182]. The combination of curcumin and quercetin has shown significant inhibition of prostate cancer cell proliferation, inducing apoptosis while modulating pathways related to inflammation, including those linked to reactive oxygen species and nitric oxide [183,184,185,186]. These synergistic effects arise from the ability of these compounds to target multiple steps within inflammatory pathways, similar to other flavonoid combinations that influence NF-κB signaling [187,188]. Additionally, curcumin and resveratrol help suppress inflammation in contrast-induced nephropathy by regulating microRNA pathways, thereby underlining their roles in managing inflammatory responses [189,190].

The combined effects of curcumin and quercetin enhance their therapeutic potential across various inflammatory conditions. The ability of natural products to promote cartilage regeneration stems primarily from their capacity to enhance MSC differentiation into chondrocytes and increase the expression of cartilage-specific extracellular matrix proteins. This process benefits from the multi-component nature of natural products, which allows them to engage multiple pathways simultaneously, thereby amplifying their therapeutic impact. For instance, traditional Chinese medicine (TCM) employs combinations of herbal ingredients that regulate various targets across different pathways, fostering synergistic actions that promote MSC differentiation and cartilage regeneration [191,192]. Furthermore, the pharmacokinetic properties of these natural products, including enhanced absorption and bioavailability, significantly contribute to their efficacy in supporting cartilage health [11,193].

Integrating natural products with conventional therapies can also enhance therapeutic outcomes by overcoming resistance mechanisms and mitigating adverse effects, as evident in cancer treatment strategies [194]. The synergistic mechanism of TCM, exemplified by the targeted combination of PepT1 and PPARα, illustrates how natural products can augment the uptake and effectiveness of active compounds, promoting cartilage regeneration [181]. Overall, incorporating natural products into therapeutic strategies presents an encouraging approach to improve MSC differentiation and cartilage regeneration through their multi-targeted and synergistic actions.

The synergistic effects of delivery systems that integrate natural products and MSCs enhance cell retention and engraftment at injury sites through various mechanisms. When used in combination, natural products have demonstrated improved therapeutic effects due to their multi-targeted approach, enhancing the bioavailability and absorption of active compounds, as shown in cardiovascular treatments [112]. This multi-component strategy is also observed in TCM, where herbal combinations like Chuanxiong Rhizome and Paeoniae Radix Rubra synergistically inhibit inflammation and oxidative stress through pathways such as PI3K/AKT and MAPK, which are crucial for cell survival and retention [195,196]. Moreover, the co-delivery of bioactive compounds can preserve and enhance radical scavenging potential, thereby sustaining the biological effects of individual components.

In cancer therapy, the combination of different therapeutic agents, including salinomycin and dasatinib, has shown improved efficacy by suppressing multiple pathways, similar to enhancing MSC retention through modulating the local microenvironment [197]. Furthermore, normalizing aberrant vasculature and endothelial activation in cancer immunotherapy can facilitate the retention and infiltration of therapeutic cells, indicating a potential mechanism for improved MSC engraftment [198]. Collectively, these findings highlight the promise of utilizing combinations of natural products and MSCs to enhance therapeutic outcomes through improved cell retention and engraftment at injury sites via synergistic mechanisms.

### 4.5. Challenges, Considerations and Future Direction

The integration of mesenchymal stromal cells (MSCs) with natural products holds significant promise for treating osteoarthritis (OA). However, several obstacles must be addressed to exploit this potential fully. One primary challenge is the inherent heterogeneity in MSC-based therapies, which complicates the evaluation of their clinical effectiveness due to variability in tissue sources, donor characteristics, and manufacturing methods [6]. This variability can impact the immunomodulatory and regenerative functions of MSCs, which are crucial for addressing both the inflammatory and degenerative components of OA. Additionally, the bioavailability and targeted application of natural products present substantial challenges. While these compounds can effectively influence inflammatory pathways such as NFκB, MAPK, and PI3K/AKT, their clinical application is often hindered by poor bioavailability and heterogeneity in disease presentation [164].

To enhance the therapeutic efficacy and clinical applicability of combining mesenchymal stem cells (MSC) and natural product therapies for osteoarthritis (OA), future research should focus on several key areas. One crucial aspect is optimizing delivery systems for MSC-derived exosomes, as these vesicles have demonstrated significant potential in modulating the joint microenvironment and promoting cartilage repair [199,200]. Integrating exosomes with biomaterials like hydrogels could improve their stability and targeted delivery, addressing current challenges in standardization and scalability [201]. Additionally, exploring the synergistic effects of natural products, such as Radix Achyranthis Bidentatae, with MSCs could enhance immune regulation and chondrocyte formation, potentially slowing OA progression [71]. Understanding the molecular pathways influenced by natural products, including NF-κB and PI3K/AKT, is essential for developing new application strategies and improving clinical outcomes. Furthermore, the role of senolytics in conjunction with MSC therapies should be investigated, as targeting senescent cells may mitigate OA progression by modulating inflammation and cellular senescence. Personalized medicine approaches, leveraging genomic insights to identify specific OA phenotypes, could also tailor treatments to individual patient needs, enhancing effectiveness and reducing adverse effects. Overall, future research should aim to integrate these diverse therapeutic strategies, focusing on mechanistic insights, delivery optimization, and personalized approaches to fully realize the potential of MSC–natural product therapies in OA management.

## 5. Mechanisms of Action

The therapeutic synergy between mesenchymal stem cells (MSCs) and natural products such as curcumin, resveratrol, quercetin, and epigallocatechin gallate (EGCG) in osteoarthritis (OA) is underpinned by their ability to modulate key cellular and molecular pathways involved in joint inflammation, cartilage degeneration, and tissue regeneration. This section highlights how natural products enhance the efficacy of MSC-based therapies, focusing on cellular processes, signaling pathways, and their combined effects on the osteoarthritic microenvironment.

### 5.1. Modulation of Inflammatory Pathways

These natural compounds are known for their anti-inflammatory and antioxidant properties, which are crucial in addressing the persistent inflammation characteristic of OA. Curcumin, for instance, has been shown to enhance the efficacy of MSC-derived exosomes in attenuating OA by modulating the miR-124/NF-kB and miR-143/ROCK1/TLR9 signaling pathways, thereby reducing inflammation and promoting chondrocyte viability [202]. Resveratrol and EGCG, along with other polyphenols, target key inflammatory pathways such as NFκB, TGFβ, and Wnt/β-catenin and execute epigenetic modifications that can alter gene expression related to OA pathophysiology [88]. These compounds can inhibit the overactivation of inflammatory cascades like NF-κB, MAPK, and PI3K/AKT, which are central to OA pathogenesis [71,140]. Moreover, the use of nano-formulated bioactive compounds, including these natural products, has been suggested to improve their bioavailability and therapeutic efficacy by modulating gut microbiota, which plays a role in systemic inflammatory responses associated with OA [203]. The integration of these natural products with MSCs not only enhances the anti-inflammatory effects but also supports cartilage homeostasis and reduces apoptosis in chondrocytes, as seen with curcumin’s ability to inhibit the PI3K-Akt-mTOR pathway [204]. Despite these promising findings, challenges such as bioavailability and precise targeting remain, necessitating further research to optimize the clinical application of these natural compounds in OA therapy. Overall, the combination of these natural products with MSCs offers a multifaceted approach to modulating inflammatory pathways in OA, potentially leading to more effective disease management strategies.

### 5.2. Reduction of Oxidative Stress

The incorporation of natural compounds alongside mesenchymal stem cells (MSCs) has demonstrated a noteworthy potential in mitigating oxidative stress [88], which is a significant challenge in stem cell-based therapies. These natural compounds are well known for their antioxidant properties [205], which include scavenging free radicals and modulating signaling pathways to enhance cellular resilience against oxidative damage. Curcumin, for instance, has been demonstrated to protect bone marrow mesenchymal stem cells (BMSCs) from hydrogen peroxide-induced oxidative stress by enhancing mitochondrial function and deactivating the Akt/Erk signaling pathways, thereby improving cell viability and reducing apoptosis [89,206]. Similarly, resveratrol and curcumin have been shown to exert synergistic effects in reducing oxidative stress and inflammation, which are crucial in maintaining the therapeutic efficacy of MSCs [69]. Quercetin and EGCG also possess direct antioxidant properties, as evidenced by their ability to scavenge free radicals in cellular assays, which can be beneficial in reducing oxidative stress when used in combination with MSCs [207]. Furthermore, the combination of these polyphenols can modulate gene expression and influence microRNA activity, thereby enhancing the antioxidant defense mechanisms at a molecular level [208]. The synergistic activity of these compounds, as observed in high-glucose-induced oxidative stress models, suggests that they can significantly decrease reactive oxygen species (ROS) production and inflammatory markers, which are critical in mitigating oxidative stress in MSC therapies [209]. Overall, the integration of these natural antioxidants with MSCs offers a multifaceted approach to enhancing therapeutic outcomes by reducing oxidative stress, thereby improving cell survival and function in various pathological conditions. However, the potential for these natural products to strengthen synergistically the regenerative capabilities of MSCs must be further explored through rigorous clinical trials. Future research should also focus on identifying the optimal combinations and dosages of these natural antioxidants to maximize their efficacy in osteoarthritis treatment.

### 5.3. Promotion of Chondrogenic Differentiation

Recent studies have investigated the integration of natural compounds such as curcumin, resveratrol, quercetin, and EGCG with mesenchymal stem cells (MSCs) as a novel strategy to enhance chondrogenic differentiation, offering significant potential for cartilage regeneration and osteoarthritis (OA) treatment. Among these, curcumin has garnered particular attention for its ability to modulate signaling pathways such as miR-124/NF-κB and miR-143/ROCK1/TLR9, which are critical in maintaining chondrocyte viability and reducing apoptosis in OA models. Remarkably, MSC-derived exosomes treated with curcumin have demonstrated the capacity to restore normal expression levels of these pathways, effectively attenuating OA progression [202]. However, this research has yet to explore additional signaling pathways that may contribute to its therapeutic outcomes. Future studies should focus on assessing curcumin’s long-term effects, its role in combination therapies, and the mechanisms underlying its epigenetic impact to expand its applications in OA management. In addition, curcumin-loaded small extracellular vesicles (sEV-CUR) derived from adipose-derived MSCs (ADMSCs) have shown superior therapeutic efficacy compared to free curcumin or ADMSC-derived sEVs. These vesicles promote chondrocyte proliferation, reduce apoptosis, and alleviate oxidative stress more effectively, making them a promising treatment option for OA. Biweekly intra-articular injections of sEV-CUR have been shown to protect cartilage and significantly reduce oxidative damage in OA models [81]. Despite these promising findings, the study does not investigate long-term effects, potential side effects, or the integration of other natural compounds such as resveratrol. Additionally, the molecular mechanisms underlying sEV-CUR’s anti-oxidative and anti-apoptotic effects remain insufficiently explored. Future research should aim to improve targeting efficiency, evaluate its pain-relief efficacy, and examine synergies with other natural compounds to harness its therapeutic potential fully.

Recent advancements in regenerative medicine continue to highlight the transformative potential of integrating natural compounds with mesenchymal stem cells (MSCs) to enhance chondrogenic differentiation, a vital process for cartilage repair and regeneration. Among these bioactive agents, resveratrol, a polyphenol, has been extensively studied for its ability to amplify the therapeutic potential of MSCs. By modulating inflammation-related cytokines, resveratrol provides a significant advantage in hyperglycemic conditions that predispose individuals to osteoarthritis (OA) [83]. Recent studies highlight the role of sustained release systems, such as resveratrol-loaded hydrogels, in maintaining localized delivery to cartilage defects, improving bioavailability and therapeutic outcomes. Resveratrol also enhances the chondrogenic differentiation of MSCs by modulating autophagy and mitochondrial dynamics, which is crucial for cells derived from metabolic syndrome conditions. Furthermore, sustained release of resveratrol has been shown to inhibit interleukin-1β-induced metalloproteinase-13 expression, thereby protecting the cartilage matrix and promoting chondrocyte differentiation [69,89]. Recent findings have highlighted the significant role resveratrol plays in modulating the behavior of mesenchymal stem cells (MSCs). These advancements focus on harnessing the bioactive compounds found in various natural sources, which have demonstrated the potential to enhance the therapeutic efficacy of injected MSCs for osteoarthritis [210]. Curcumin, another natural compound, has been reported to synergize with resveratrol to stimulate the MAPK signaling pathway, which is vital for chondrocyte differentiation and survival. It also suppresses the nuclear factor-κB pathway, thereby promoting chondrogenic differentiation in a high-density co-culture microenvironment [85]. The combination of these natural products with MSCs not only enhances the differentiation potential but also addresses the inflammatory and oxidative stress challenges that often impede effective cartilage regeneration. While the specific effects of quercetin and EGCG in combination with MSCs were not detailed in the provided contexts, the synergistic effects of curcumin and resveratrol highlight the potential of using natural compounds to enhance MSC-based therapies for cartilage repair. Overall, these findings suggest that integrating natural products with MSCs could significantly improve the outcomes of regenerative treatments for osteochondral injuries and degenerative joint diseases like OA.

### 5.4. Enhanced Delivery via Biomaterials

These natural polyphenols are known for their antioxidant, anti-inflammatory, and anticancer properties, which are crucial in biomedical applications, including drug delivery and tissue engineering. For instance, resveratrol has been shown to improve the survival, self-renewal, and differentiation of MSCs, thereby enhancing their therapeutic effects in regenerative medicine [53]. Similarly, resveratrol preincubation with MSCs enhances their migration and survival in neurodegenerative disease models, such as Alzheimer’s, by modulating neuroinflammation and improving cell homing [211]. The use of nanoparticles and hydrogels as delivery systems further augments the stability and bioavailability of these polyphenols. For example, curcumin and resveratrol co-delivered via a hydrogel/nanoparticle system demonstrated significant anti-inflammatory effects in a spinal cord injury model [212]. Additionally, quercetin and resveratrol encapsulated in targeted nanoparticles improved cellular uptake and stability, enhancing their therapeutic potential [213]. The formation of phytosomes with phospholipids also increases the bioavailability of these polyphenols, facilitating their penetration into cells [214]. Moreover, the use of nanotechnology, such as sericin self-assembling nanoparticles, allows for a controlled release of polyphenols, promoting MSC metabolic activity and protecting them from oxidative stress [215]. These advanced delivery systems not only improve the pharmacokinetic properties of natural products but also enhance the regenerative capabilities of MSCs, making them promising candidates for treating various diseases and injuries.

### 5.5. Immune Modulation

The combination of natural products such as curcumin, resveratrol, quercetin, and epigallocatechin-3-gallate (EGCG) with mesenchymal stem cells (MSCs) has shown promising effects on immune modulation. These natural compounds are known for their anti-inflammatory and immunomodulatory properties, which can enhance the therapeutic potential of MSCs in various pathological conditions. Curcumin, quercetin, and resveratrol, when used together, have been shown to modulate the tumor microenvironment by increasing T cell recruitment and reducing immunosuppressive cell populations, thereby tipping the immune balance towards activation [79]. Quercetin, in particular, has been demonstrated to enhance the immunoregulatory effects of MSCs by modulating pathways such as Akt/IκB and Toll-like receptor-3, which are crucial for immune response regulation [216]. EGCG, another potent polyphenol, has been noted for its role in modulating neuroimmune diseases, suggesting its potential to enhance MSC-based therapies for immune-related disorders and improve the therapeutic efficacy of mesenchymal stem cells (MSCs) in treating osteoarthritis. Recent studies have indicated that EGCG can improve the viability and functionality of MSCs, potentially leading to better patient outcomes [63,217,218]. The combination of these natural products with MSCs can potentially improve the modulation of inflammatory phenotypes and immune responses, offering a novel therapeutic avenue for autoimmune and inflammatory diseases. Furthermore, these compounds have been shown to influence various immune pathways, such as NF-κB and IL-6/JAK/STAT, which are critical in controlling immune responses and inflammation [219,220,221]. Overall, the integration of these natural products with MSCs could provide a synergistic effect, enhancing the immunomodulatory capacity of MSCs and offering a promising strategy for treating immune-related diseases.

### 5.6. Future Perspectives

Understanding the mechanisms of action underlying mesenchymal stem cell (MSC) and natural product therapies is crucial for optimizing their application in osteoarthritis (OA) treatment. MSCs have shown promise due to their regenerative and immunomodulatory properties, which include promoting chondrocyte regeneration, inhibiting extracellular matrix degradation, and attenuating inflammation induced by macrophages [222,223]. Exosomes derived from MSCs further enhance these effects by delivering bioactive molecules that modulate the joint microenvironment, reduce inflammation, and promote cartilage repair. However, challenges such as low yield, weak activity, and inefficient targeting of exosomes need to be addressed through engineering approaches and biomaterial integration to improve their therapeutic efficacy. On the other hand, natural products (NPs) offer a complementary approach by modulating key inflammatory pathways like NF-κB, MAPK, and PI3K/AKT, which are central to OA pathogenesis. Despite their potential, the clinical translation of NPs is hindered by issues of bioavailability and precise targeting [70,224]. Future research should focus on optimizing drug delivery systems for both MSC-derived exosomes and NPs, exploring personalized medicine approaches, and conducting comparative effectiveness studies to realize their full potential in OA treatment. Additionally, understanding the interaction between MSCs and the OA microenvironment, including inflammatory and senescence factors, is essential for enhancing the efficacy of MSC therapies. By addressing these research gaps, it may be possible to develop more effective and safer therapeutic strategies for managing OA, ultimately improving patient outcomes.

### 5.7. Perspectives on Clinical Practice and Translation

The integration of mesenchymal stem cells (MSCs) with natural products such as curcumin, resveratrol, quercetin, and epigallocatechin gallate (EGCG) presents a promising strategy for osteoarthritis (OA) treatment. These therapies could revolutionize clinical approaches by offering disease-modifying solutions that not only reduce inflammation but also promote cartilage repair and regeneration [225]. As natural products can enhance MSC viability, differentiation, and immune modulation, their clinical application could significantly alter the current landscape of OA management [226]. Currently, OA treatments primarily focus on symptom management through pharmacological therapies, such as NSAIDs and corticosteroid injections, or invasive options such as joint replacement surgery [227]. However, these strategies fail to address the underlying degenerative processes and often come with long-term side effects [228]. The introduction of MSCs combined with natural products could provide a regenerative alternative that targets the root causes of OA, offering the potential to slow disease progression, reduce inflammation, and repair cartilage.

In clinical practice, the incorporation of MSCs with natural products could complement or even replace some current therapies. For example, MSCs preconditioned with bioactive compounds like curcumin or resveratrol could improve patient outcomes by enhancing the regenerative capacity of the stem cells, reducing the need for pain management strategies like NSAIDs or corticosteroid injections [57]. This shift could lead to a more sustainable approach to OA care, with less reliance on drugs that come with significant side effects. Despite their promising potential, several challenges must be addressed to bring MSC–natural product therapies into routine clinical practice. One of the most significant barriers is bioavailability. Natural products like curcumin and resveratrol are known for their potent therapeutic effects, but their clinical use is often limited by poor bioavailability and rapid metabolism [113]. Strategies to improve the delivery of these compounds, such as nanoparticle formulations or hydrogel-based systems, are promising, but they need further optimization for clinical application.

Another critical challenge is the variability in MSC sources and protocols. MSCs can be derived from various tissues, such as bone marrow, adipose tissue, and umbilical cord blood, and there is significant variation in how these cells are isolated, cultured, and administered [229]. This variability can lead to inconsistent therapeutic outcomes, making it difficult to standardize treatment protocols. Furthermore, the regulatory landscape remains complex, as the clinical translation of MSC therapies requires rigorous safety evaluations and long-term follow-up studies to ensure their efficacy and safety. Additionally, the heterogeneity of OA across patient populations complicates the development of a universal therapeutic approach. Factors such as age, comorbidities, disease severity, and genetic predisposition can all affect how a patient responds to MSC-based therapies. This variability requires personalized treatment strategies, where patient-specific factors are considered to tailor the therapy to individual needs.

To address these challenges, future research must focus on optimizing delivery systems for natural products, such as the use of nanoparticle encapsulation or sustained-release hydrogels. These strategies would enhance the stability and bioavailability of bioactive compounds, ensuring they reach therapeutic concentrations at the target site. Moreover, standardization of MSC protocols is crucial to ensure consistency in the production and administration of stem cells. Establishing standardized methods for MSC isolation, culture, and characterization would streamline clinical application and improve reproducibility across clinical trials. To overcome the regulatory hurdles, it is essential to establish clear guidelines for the clinical use of MSC–natural product therapies. Regulatory bodies must work alongside researchers to create frameworks that address the unique challenges posed by stem cell-based therapies, ensuring that treatments are both safe and effective. Additionally, incorporating advanced biomarkers to better understand patient-specific responses could pave the way for more effective, personalized treatments. By integrating biomarkers into clinical trials, we could identify which patients are most likely to benefit from MSC therapies combined with natural products, thereby improving patient outcomes and minimizing the risk of adverse effects.

Finally, collaborative efforts between academia, industry, and regulatory agencies are needed to accelerate the development of MSC–natural product therapies. Multi-center clinical trials and robust safety data will be essential for demonstrating the efficacy and long-term safety of these therapies. As these therapies evolve, there will also be an opportunity to explore combination therapies that integrate MSCs, natural products, and other regenerative medicine strategies, creating a more comprehensive approach to OA treatment.

## 6. Challenges and Considerations

The combination of mesenchymal stem cells (MSCs) and natural products offers significant possibilities for osteoarthritis (OA) treatment; nonetheless, several obstacles must be addressed to enable the shift of these therapies from research environments to clinical use. This section highlights key challenges, including the necessity for standardization, concerns related to bioavailability, safety issues, regulatory hurdles, and possible approaches to tackle these challenges effectively.

### 6.1. Standardization and Quality Control

A primary obstacle is the standardization of MSC-derived exosome production and characterization, which is crucial for ensuring consistent therapeutic outcomes. Exosomes, which are cell-free vesicles derived from MSCs, have shown potential in modulating the joint microenvironment, reducing inflammation, and promoting cartilage repair [230,231]. However, the scalability of production and a comprehensive understanding of their mechanisms remain significant hurdles. Additionally, the integration of MSCs with natural products like Radix Achyranthis Bidentatae (AB) has shown the potential to enhance joint function and reduce cartilage apoptosis [232]. Yet, the precise mechanisms and optimal combinations require further exploration. The use of biomaterials, such as hydrogels and magnetic polysaccharide microcarriers, has been proposed to improve the delivery and efficacy of MSC-derived exosomes [233,234,235]. Still, these approaches also face challenges related to efficient enrichment and targeted delivery. Furthermore, clinical trials have demonstrated variability in patient responses to MSC therapies [6], highlighting the need for personalized treatment strategies and a deeper understanding of patient-specific factors that influence therapeutic efficacy. Addressing these challenges through rigorous standardization, mechanistic studies, and personalized approaches will be essential for the successful clinical translation of MSC and natural product-based therapies for OA.

### 6.2. Bioavailability and Delivery

One significant obstacle is the efficient delivery and retention of therapeutic agents within the joint space, as conventional drug delivery systems often suffer from rapid clearance and poor bioavailability, limiting their effectiveness [236]. Innovative delivery systems, such as hydrogels and lipid-based drug delivery systems, have been explored to enhance the stability and targeted delivery of MSCs and their derivatives, such as exosomes, to the affected areas. Hydrogels, for instance, provide a biocompatible scaffold that supports MSC viability and differentiation, which is crucial for effective cartilage repair [237]. Additionally, magnetic polysaccharide microcarriers and DNA supramolecular hydrogels have shown the potential to protect MSCs from shear forces and enhance their therapeutic efficacy in OA environments [238,239]. Furthermore, the combination of MSCs with natural products like Radix Achyranthis Bidentatae and stigmasterol has demonstrated synergistic effects in promoting cartilage repair and reducing inflammation [71]. However, the delivery of these combinations requires further optimization. The development of multifunctional delivery systems that can provide sustained release and targeted delivery of therapeutic agents is crucial for improving treatment outcomes and minimizing systemic exposure. Addressing these bioavailability and delivery challenges through advanced biomaterial engineering and innovative delivery strategies is essential for the successful clinical translation of MSC-based therapies for OA.

### 6.3. Safety and Potential Side Effects

Several challenges must be addressed to transition these therapies from research to clinical practice, particularly concerning safety and potential side effects. MSC-derived exosomes (MSC-Exos) have shown potential in modulating the joint microenvironment and promoting cartilage repair, but their clinical application is hindered by the need for standardized production methods and long-term safety assessments [201,240]. The safety of MSC therapies has been explored in various studies, with intra-articular injections of adipose-derived MSCs (ADMSCs) demonstrating no severe treatment-related adverse events in both pilot and larger-scale trials, suggesting a favorable safety profile [161,235,236]. However, the variability in patient responses and the lack of large-scale, controlled studies highlight the need for further research to confirm these findings and understand the mechanisms underlying differential responses. Additionally, the integration of MSCs with natural products like Radix Achyranthis Bidentatae (AB) offers potential benefits in immune regulation and cartilage protection [232]. Still, the safety of such combinations requires thorough investigation. The development of delivery systems, such as magnetic polysaccharide microcarriers and gelatin methacryloyl hydrogels, aims to enhance the therapeutic efficacy and safety of MSC-based treatments by ensuring targeted delivery and sustained release [201]. Yet, these innovations also necessitate rigorous safety evaluations. Overall, while MSC-based therapies for OA show promise, addressing safety concerns through comprehensive clinical trials and mechanistic studies is crucial for their successful clinical translation.

### 6.4. Regulatory Hurdles

One of the primary obstacles faced in the field of regenerative medicine is the regulatory hurdles associated with MSC-based therapies. These hurdles include the urgent need for standardized protocols for the isolation, characterization, and production of mesenchymal stem cells (MSCs), which are vital to ensure consistent quality, efficacy, and safety across various studies and applications [6,91,238].

The significant variability in methodologies and outcomes observed in clinical trials further complicates the regulatory landscape, primarily because there exists a noticeable lack of consensus among researchers and regulatory bodies on the most effective MSC source, optimal dosage, and appropriate delivery methods for these therapies [6]. Moreover, the long-term safety and efficacy of MSC therapies remain under scrutiny, creating a necessity for rigorous clinical trials that can sufficiently establish their therapeutic potential while meticulously addressing any possible adverse effects that may arise during treatment [42,239]. In addition, the integration of natural products, such as the well-known Radix Achyranthis Bidentatae, with mesenchymal stem cells (MSCs) warrants careful and systematic evaluation [11,232]. This thorough investigation is necessary to fully comprehend their potential synergistic effects and optimize the corresponding treatment protocols aimed at achieving better patient outcomes. Furthermore, the ongoing development of innovative delivery systems, such as cell–tissue matchmaking nanoconstructs, highlights the critical need for advanced technologies specifically designed to enhance MSC implantation and integration within the target tissues [240]. These advancements are essential for significantly improving overall therapeutic outcomes in various clinical settings, where the merging of science and medicine can lead to groundbreaking innovations that benefit patients. The collaboration between natural therapies and modern medical practices has the potential to revolutionize treatment options and contribute to enhanced healing processes, paving the way for more effective interventions. Addressing these ever-evolving regulatory and technical challenges is not just a necessity but an essential prerequisite for the successful clinical translation of mesenchymal stem cell (MSC) therapies and natural product-based treatments, specifically for osteoarthritis (OA). This concerted and dedicated effort ultimately seeks to provide a wider array of effective and regenerative treatment options for patients who suffer from painful degenerative joint diseases, intending to significantly improve their overall quality of life through innovative and groundbreaking medical advancements that could greatly enhance their everyday experiences and functional abilities.

Despite promising findings, several challenges must be addressed before MSC–natural product therapies can be fully integrated into clinical practice. This section discusses key limitations and research priorities for future studies.

## 7. Challenges, Limitations, and Conflicting Evidence

Despite promising preclinical and early clinical findings, significant challenges and limitations remain in integrating natural products with MSC-based therapies for osteoarthritis (OA). This section explores four key areas of concern: bioavailability issues, variability in MSC sources, conflicting clinical data, and the need for standardized treatment protocols.

### 7.1. Bioavailability and Stability Challenges

One of the primary obstacles to using curcumin, resveratrol, quercetin, and EGCG in MSC therapy is their poor bioavailability. These compounds face issues such as low solubility, rapid metabolism, and inefficient systemic absorption. For instance, curcumin undergoes rapid degradation and hepatic metabolism, limiting its therapeutic efficacy in vivo [67]. Similarly, EGCG, despite its potent antioxidant properties, has a short half-life and low stability in physiological conditions, reducing its bioactive potential [63]. Resveratrol, although capable of enhancing MSC survival, is poorly absorbed in the gastrointestinal tract [68].

To address these challenges, recent strategies have focused on improving the stability and absorption of these bioactive compounds. Nanoparticle encapsulation, liposomal formulations, and hydrogel-based delivery systems have shown promise in enhancing bioavailability [52,75]. Future research should prioritize optimizing nano-formulations for controlled release in the joint microenvironment and conducting comparative studies on different bioavailability enhancement techniques.

### 7.2. Variability in MSC Sources and Treatment Efficacy

The therapeutic potential of MSCs varies depending on factors such as tissue source, donor age, and culture conditions. This variability presents a significant challenge in standardizing treatments. Bone marrow-derived MSCs (BM-MSCs) are commonly used but have lower proliferation rates in aged individuals. Adipose-derived MSCs (AD-MSCs) are more abundant and accessible, but their immunomodulatory potential is less well studied in OA [241,242]. Umbilical cord-derived MSCs (UC-MSCs) show high proliferative potential but are less frequently tested in clinical OA settings.

Furthermore, differences in MSC preparation protocols can lead to inconsistencies in therapeutic outcomes. To address these issues, future studies should compare different MSC sources under the same experimental conditions to determine which type offers optimal regenerative potential for OA treatment. Additionally, establishing standardized MSC culture and expansion protocols is crucial to ensure reproducibility in clinical trials.

### 7.3. Conflicting Clinical Data on MSC + Natural Product Therapies

While preclinical studies demonstrate strong evidence of natural products enhancing MSC function, clinical translation remains inconsistent. Some clinical trials report significant pain reduction and functional improvements in OA patients treated with MSCs and bioactive compounds [69]. However, other studies show no significant cartilage regeneration, suggesting that the therapeutic window for MSC + natural product therapy may be patient-specific [85].

These conflicting results can be attributed to differences in patient age, OA severity, MSC dosage, and natural product administration. To resolve these discrepancies, larger, multi-center randomized controlled trials (RCTs) are needed to determine long-term efficacy and optimal dosing regimens. Additionally, developing biomarker-based patient selection criteria could help identify individuals who will benefit most from MSC + natural product therapy.

### 7.4. Lack of Regulatory Guidelines and Standardized Protocols

The absence of a standardized regulatory framework for combining natural products with MSC-based cell therapies presents another significant challenge. Many natural compounds are classified as dietary supplements rather than pharmaceuticals, leading to variability in quality control [74]. While MSC therapies require strict Good Manufacturing Practice (GMP) compliance, the co-administration of bioactive compounds is not yet fully regulated in most countries.

To address this issue, regulatory agencies should develop clear guidelines on combining stem cell-based therapies with bioactive compounds. Future studies should also focus on standardizing treatment regimens, including optimal dosing, frequency, and delivery methods for combined MSC + natural product therapy.

### 7.5. Future Research Priorities

In conclusion, while the integration of natural products into MSC-based therapies for OA is a promising frontier, several key research areas must be addressed before clinical translation is fully realized. These include improving bioavailability through advanced formulations, comparing different MSC sources in head-to-head trials, conducting larger-scale human trials with ≥200 patients to confirm long-term benefits, and establishing clear regulatory pathways for combined therapies. By addressing these challenges, researchers can pave the way for more effective and standardized treatments for osteoarthritis.

## 8. Conclusions and Future Directions

The integration of natural products with mesenchymal stem cell (MSC) therapies represents a promising strategy for enhancing cartilage regeneration, reducing inflammation, and improving clinical outcomes in osteoarthritis (OA). This review highlights the potential of curcumin, resveratrol, quercetin, and EGCG to modulate key inflammatory pathways (NF-κB, MAPK, PI3K/AKT), enhance MSC viability, and promote chondrogenic differentiation. Despite encouraging preclinical and early clinical findings, significant challenges remain before these therapies can be widely adopted in clinical settings.

### 8.1. Research Priorities for Advancing MSC + Natural Product Therapies

Future research should focus on several key areas to optimize MSC–natural product therapy for OA. First, improving the bioavailability and delivery systems of natural compounds is crucial, as many have poor solubility, rapid metabolism, and low systemic absorption. Studies should explore nanoparticle-based drug delivery, hydrogel formulations, and lipid-based carriers to enhance controlled release and joint retention.

Second, it is essential to compare different MSC sources and standardize therapy protocols. Bone marrow-derived MSCs (BM-MSCs), adipose-derived MSCs (AD-MSCs), and umbilical cord MSCs (UC-MSCs) exhibit varying proliferation rates, immunomodulatory capacities, and chondrogenic potential. Direct comparisons under identical conditions will help determine the most effective MSC type for OA treatment.

Third, conducting large-scale, randomized controlled trials (RCTs) is necessary to confirm the safety, efficacy, and long-term benefits of MSC–natural product therapies. Current clinical trials are often small-scale (≤50 patients) and lack long-term follow-up data. Future multi-center RCTs with larger patient cohorts (≥200 patients) will provide more robust evidence.

Fourth, establishing mechanism-based biomarkers for patient selection will improve treatment outcomes. As not all OA patients respond equally to MSC therapy, incorporating biomarkers such as inflammatory cytokine profiles and genetic markers can help identify patients who will benefit most from MSC–natural product therapy.

Lastly, addressing regulatory and manufacturing challenges is crucial for clinical translation. Standardizing protocols for MSC isolation, expansion, and combination with natural products will ensure reproducibility and facilitate regulatory approval.

### 8.2. Clinical Translation: Roadmap for the Next 5–10 Years

Given the advancements in stem cell therapy, biomaterial-based drug delivery, and regenerative medicine, the clinical integration of MSC and natural product therapies for OA is feasible within the next decade. A proposed roadmap for clinical translation includes three main phases:

Preclinical refinement (0–2 years): This phase focuses on optimizing bioavailability solutions, comparing MSC sources, and conducting high-throughput screening to identify the most effective MSC–natural product combinations.Small-scale clinical trials (2–5 years): During this period, phase I/II clinical trials will assess safety, dosing, and preliminary efficacy. Simultaneously, standardized MSC culture and expansion protocols will be developed, and collaboration with regulatory agencies will define safety standards for combined therapies.Large-scale clinical adoption (5–10 years): The final phase involves conducting multi-center phase III trials, establishing personalized treatment strategies based on patient biomarkers, and achieving regulatory approval and commercialization of MSC–natural product combination therapy for OA.

### 8.3. The Future of MSC + Natural Product Therapy for OA

The integration of natural products into MSC-based regenerative therapies represents a paradigm shift in OA treatment. By combining MSCs’ regenerative potential with the pharmacological benefits of bioactive compounds, these therapies could offer a disease-modifying approach rather than just symptomatic relief. However, to transition from promising research to clinical reality, future studies must prioritize bioavailability optimization, MSC standardization, and large-scale clinical trials. With continued research and development, MSC + natural product therapies have the potential to revolutionize OA treatment, offering hope for millions of patients worldwide.

## Figures and Tables

**Figure 1 cimb-47-00445-f001:**
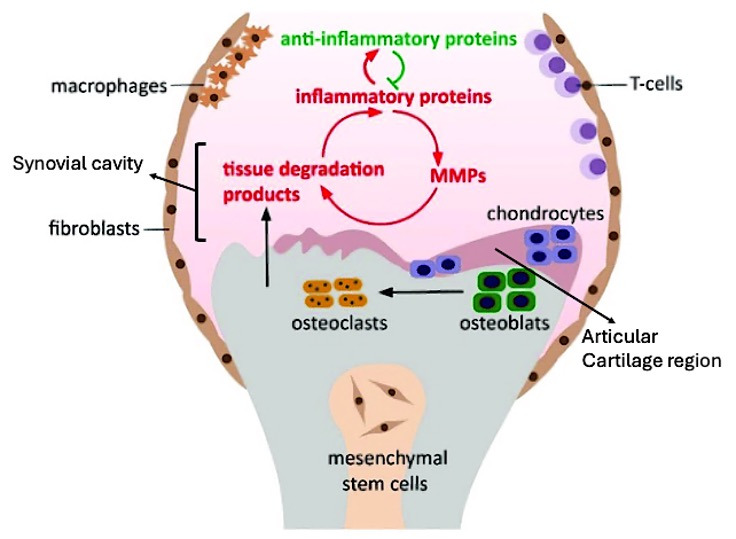
Schematic representation of osteoarthritis (OA) pathology involving various cellular and molecular components. The diagram highlights the “Articular Cartilage Region” (purple section in the upper middle), which consists of chondrocytes, that are responsible for maintaining cartilage structure. Surrounding this region, the “Synovial Cavity” (pink area), the space above chondrocytes, that facilitates fluid exchange and mediates interactions between inflammatory proteins, anti-inflammatory agents, and tissue degradation processes. Mesenchymal stem cells (MSCs) are centrally located, surrounded by macrophages, fibroblasts, and T-cells, indicating an inflammatory milieu. The interplay between inflammatory and anti-inflammatory proteins, depicted by green and red arrows respectively, stimulates matrix metalloproteinases (MMPs) and generates tissue degradation products, thereby exacerbating the inflammatory cycle. Osteoclasts and osteoblasts are involved in bone remodeling, while chondrocytes contribute to cartilage repair. This dynamic balance between inflammation and tissue repair is a hallmark of OA progression.

**Figure 2 cimb-47-00445-f002:**
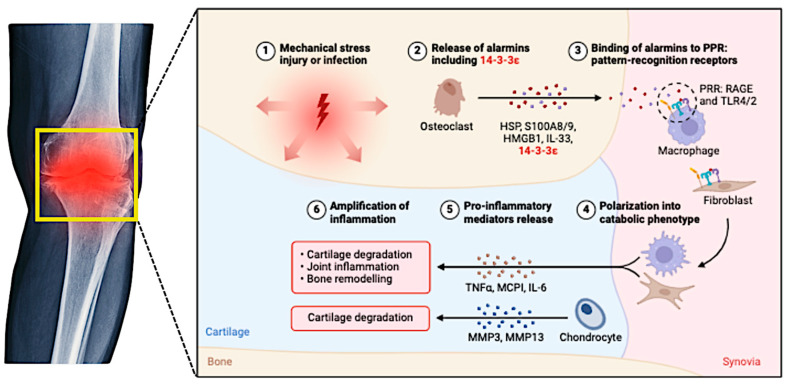
This figure illustrates the complex inflammatory pathways involved in the pathogenesis and progression of osteoarthritis (OA).

**Table 1 cimb-47-00445-t001:** Current and potential treatment for osteoarthritis (OA).

Treatment Category	Specific Treatments	Description	Effects
Non-Pharmacological	Regular physical activity	Promotes mobility and joint function	Aids weight reduction and overall health improvement
	Lifestyle modifications	Diet and other health-related changes	Critical for patients with obesity or poor nutrition
	Weight management	Lifestyle changes, including diet and exercise	Reduces joint stress and improves overall health
	Physical therapy	Exercises to improve joint mobility and muscle strength	Alleviates pain and enhances functional capacity
Pharmacological	Glucosamine and chondroitin	Supplements for cartilage health	Clinical results are inconsistent; generally not recommended
	Steroid injections	Corticosteroid injections for acute pain relief	Short-term pain relief and long-term use should be monitored
	Duloxetine	Antidepressant approved for chronic pain in knee OA	Offers better relief in combination therapies compared to NSAIDs alone
	Opioids/narcotics	Strong analgesics for severe cases	Effective but associated with addiction and adverse effects
	Paracetamol	Analgesic for mild to moderate pain relief	Safe but less effective than NSAIDs
	Nonsteroidal anti-inflammatory drugs	Cyclooxygenase inhibitors for pain relief	Effective pain relief but requires caution due to gastrointestinal side effects.
	Viscosupplementation	Hyaluronic acid injection for joint lubrication	Offers potential chondroprotective benefits; effectiveness is debated
Surgical	Arthroscopy	Minor surgical adjustment within the joint	Less invasive, but the long-term effectiveness is debated
	Microfracture surgery	Stimulates growth of new cartilage	Standard due to low cost, it results in less-durable fibrocartilage
	Total joint arthroplasty	Full joint replacement surgery	Practical in advanced cases; significant improvements in quality of life
Emerging Treatments	Platelet-rich plasma (PRP)	Concentrated platelets for tissue regeneration	Aids in inflammatory response modulation
	Gene therapy	Introducing therapeutic genes directly to the joint	Potential for long-term effects and slowing degenerative processes
	Small molecule inhibitors	Target specific pathways of joint inflammation	Promising disease modification and symptomatic relief
	Stem cell therapy	Mesenchymal stem cells for tissue regeneration	Potentially modulates inflammation and promotes regeneration

**Table 2 cimb-47-00445-t002:** Summary of key studies on natural products enhancing MSC therapy for osteoarthritis.

Authors & Year	Study Type	Natural Product	MSC Source	Main Findings	Limitations
[67]	In vitro	Curcumin Dosage: 500–2000 mg/day; frequency: daily, divided doses	Bone marrow MSCs	Enhances chondrogenic differentiation and reduces oxidative stress	Lacks in vivo validation
[68]	Animal (OA model)	ResveratrolDosage: 150–500 mg/day (up to 1000 mg); frequency: once daily or divided doses	Adipose-derived MSCs	Reduces cartilage degradation and inflammation	No human trial evidence
[69]	Clinical trial	Resveratrol Dosage: 150–500 mg/day (up to 1000 mg); frequency: once daily or divided doses	Bone marrow MSCs	Improved pain and function in knee OA patients	Small sample size (n = 30)
[70]	In vitro	Quercetin Dosage: 500–1000 mg/day; frequency: divided doses	Umbilical cord MSCs	Anti-inflammatory effects, reduces MMP expression	No long-term study
[71]	In vivo (rat model)	EGCG Dosage: 400–800 mg/day; frequency: once or twice daily	Bone marrow MSCs	Protects cartilage from inflammatory damage	Bioavailability concerns
[72]	In vivo (mouse model)	Resveratrol Dosage: 40 mg/kg/day; once daily, 4 weeks. s	Bone marroe MSCs	Enhances osteogenic differentiation by activating SIRT1	No in vivo data
[73]	In vitro	Quercetin Dosage: 500–1000 mg/day; frequency: divided doses	Bone marrow MSCs	Modulates macrophage polarization, reduces inflammatory cytokines	Needs validation in animal models
[63]	In vivo (OA model)	EGCG Dosage: 400–800 mg/day; frequency: once or twice daily	Bone marrow MSCs	Protects MSCs from apoptosis and oxidative stress	No clinical translation yet
[74]	In vitro and in vivo	Curcumin + resveratrol (no information)	Bone marrow MSCs	Synergistic effect in reducing TNF-α-induced inflammation	Dosage standardization needed
[56]	In vitro	Quercetin + EGCG (no information)	Bone marrow MSCs	Enhances MSC survival under oxidative stress	No human trial conducted
[52]	Animal (OA model)	Curcumin-loaded nanoparticles Dosage: 10.32% drug loading; frequency: sustained release over 10 days	Bone marrow MSCs	Improves MSC retention and differentiation in OA joints	Requires more safety studies for clinical use
[75]	In vivo (OA model)	EGCG-loaded hydrogelsDosage: 10 mg/kg; frequency: sustained release	Adipose-derived MSCs	Increases MSC viability and cartilage regeneration	Lacks clinical trial data

The following sections provide a detailed analysis of how individual natural products—such as curcumin, resveratrol, quercetin, and EGCG—enhance MSC function, modulate inflammatory pathways, and improve therapeutic outcomes.

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
