# Peer review of "Synergistic Effects of Natural Products and Mesenchymal Stem Cells in Osteoarthritis Treatment: A Narrative Review"

_cimb, 2025, doi:10.3390/cimb47060445_

Round 1
Reviewer 1 Report
Comments and Suggestions for Authors
In this review, Alfaqeh at al. briefly summarized the molecular and cellular mechanisms involved in osteoarthritis (OA), a joint disease often referred to as “a wound that doesn't heal”. The authors showed that mesenchymal stem cells (MSCs) are an attractive candidate for OA treatment, although some problems remain. In the search for innovative therapies, recent studies have shown that natural products from plants and microorganisms in combination with MSCs could be a promising treatment strategy for OA. Several preclinical studies have shown that curcumin and quercetin, for instance, reduce inflammation and oxidative stress, thereby improving stem cell survival and function. In addition, these products promote the differentiation of stem cells into chondrocytes while at the same time suppressing matrix metalloproteinases, which is essential for maintaining articular cartilage integrity. These strategies could eventually lead to improved clinical outcomes for osteoarthritis patients. While the results are encouraging, there are still some research gaps (e.g. dosages, combinations and bioavailability of natural products, targeted delivery systems, safety and long-term effects) that need to be addressed before MSC + natural products-based therapies can be incorporated into clinical practice. Furthermore, personalized treatment plans should be developed to address individual needs.
This literature review is well written and entertaining to read. It summarizes a large amount of data and outlines possible strategies for the future treatment of osteoarthritis. I have just a few minor comments.
Minor comments
Fig. 1 The legend can be improved. E.g. Schematic representation of a synovial joint showing the various cellular and molecular components that lead to osteoarthritis (OA). Moreover, the authors should label the articular cartilage and the synovial cavity. They should also clarify that the fibroblast layer represents the synovial membrane.
Page (P)3, Line (L)93: What do the authors mean by “deformities”? Deformities of what? The bones?
P4, L136: ….“osteophyte formation of osteophytes”. What does it mean? The authors can delete “of osteophytes”.
P6, L206: It should read “Figure 2”.
P13, L457: I think it should read “Curcumin’s”
P14, L518 - 521: I do not understand the meaning of this sentence. It should be reworded.
Reviewer 2 Report
Comments and Suggestions for Authors
This review integrates the latest findings on how natural products can significantly improve the efficacy of injected MSCs in the treatment of OA, exploring the mechanism of action as well as the synergistic effects in combination therapies. This is a comprehensive and attractive manuscript. However, several issues remain to be resolved before this manuscript can be accepted for publication.
1.It is recommended that specific dosage regimens (concentration and frequency, etc.) for natural product treatments be added to the Tables.
2.It is recommended that the authors appropriately add the specific implications of these natural products for clinical practice and their potential for clinical translation in the Perspectivessection. In addition, a more comprehensive discussion of the challenges and limitations that may be encountered in translating these natural product-related discoveries into clinical practice should be included.
